# Missing data in amortized simulation-based neural posterior estimation

**Zijian Wang**[1], **Jan Hasenauer**[1,2]*, **Yannik Schälte**[1,2,3]*

**1** University of Bonn, Life and Medical Sciences Institute, Bonn, Germany, **2** Helmholtz Center Munich, Computational Health Center, Neuherberg, Germany, **3** Technical University Munich, Center for Mathematics, Garching, Germany

* jan.hasenauer@uni-bonn.de (JH); yannik.schaelte@uni-bonn.de (YS)

## Abstract

Amortized simulation-based neural posterior estimation provides a novel machine learning based approach for solving parameter estimation problems. It has been shown to be computationally efficient and able to handle complex models and data sets. Yet, the available approach cannot handle the in experimental studies ubiquitous case of missing data, and might provide incorrect posterior estimates. In this work, we discuss various ways of encoding missing data and integrate them into the training and inference process. We implement the approaches in the BayesFlow methodology, an amortized estimation framework based on invertible neural networks, and evaluate their performance on multiple test problems. We find that an approach in which the data vector is augmented with binary indicators of presence or absence of values performs the most robustly. Indeed, it improved the performance also for the simpler problem of data sets with variable length. Accordingly, we demonstrate that amortized simulation-based inference approaches are applicable even with missing data, and we provide a guideline for their handling, which is relevant for a broad spectrum of applications.

## Author summary

In biomedical research, mechanistic models describe dynamic processes, yet inferring their underlying parameters can often be challenging. Bayesian statistics provides an established framework for this by integrating prior knowledge with observed data, and naturally enables uncertainty quantification as a distribution of parameter values is returned. However, classical case-based methods for Bayesian inference can be computationally expensive, particularly when the same model needs to be fitted to different data sets. Recently, deep-learning-based approaches have been developed to streamline the inference procedure, allowing the upfront training cost to amortize when applied to multiple data sets. In this manuscript, we explore approaches to extend the setup to data sets with missing data. In summary, an encoding scheme which exploits data augmentation with binary indicators of presence or absence performs the most robustly across different test problems.

**Data Availability Statement:** The full code underlying this study can be found at https://github.com/emune-dev/Data-missingness-paper, a snapshot of code and data is available on Zenodo at https://doi.org/10.5281/zenodo.7515458.

**Funding:** This work was supported by the German Federal Ministry of Education and Research (BMBF) (EMUNE/031L0293C and FitMultiCell/031L0159C) and the German Research Foundation (DFG) under Germany's Excellence Strategy (EXC 2047 - 390685813 and EXC 2151 – 390873048) and the Schlegel Professorship for J.H.. Y.S. acknowledges financial support by the Joachim Herz Stiftung. The funders had no role in study design, data collection and analysis, decision to publish, or preparation of the manuscript.

**Competing interests:** The authors have declared that no competing interests exist.

# 1 Introduction

Mechanistic models are used to describe and understand dynamical systems in a variety of research fields ranging from life and physical sciences to economics [1, 2]. Commonly, these models depend on unknown parameters, which can be estimated by assessing the likelihood of observed data given parameters [3, 4]. Classical parameter estimation methods (e.g. optimization, Markov-chain Monte-Carlo, approximate Bayesian computation [4, 5]) are case-based. That is, they work on the level of individual data sets, such that the entire computationally expensive inference procedure needs to be repeated for every new data set.

However, often the same structural model is fitted to different data sets with potentially different parameters, e.g. to describe experiments under different stimuli, epidemic dynamics in different communities, or treatment response for different patients. In such cases, *amortized* inference methods are of interest. These first learn a mapping from synthetic data sets to e.g. likelihood or posterior distribution, which can subsequently be cheaply queried for many observed data sets [6, 7]. A successful method, which is particularly applicable for the study of time series models, is BayesFlow. BayesFlow uses conditional invertible neural networks (cINN) to learn, conditioned on data, a reversible transformation from parameters to a tractable latent space, and has been shown to be superior to alternative approaches capable of amortized simulation-based inference, as well as to case-based methods when facing multiple data sets [8].

A problem persistent in many research areas is that data are incomplete, i.e. parts of the entries are missing. There are many possible reasons, including incomplete entry, data loss, device malfunction, or study participant non-response. Further, in clinical studies measurements are often not taken at the exact time intervals or for different time spans, across patients or participants. There exist various missingness mechanisms as well as strategies to deal with them, e.g. by deletion or imputation [9]. In case-based inference, at its simplest, a likelihood or cost function can be formulated based on only the available data. However, amortized inference typically requires inputs to be of consistent structure and size across samples, and needs to know what entries are available, in order to learn the underlying data-parameter relationships correctly. For the specific case of data sets of different sizes, BayesFlow already allows to use summary networks yielding a fixed-size representation. However, this mechanism permits only e.g. time series of different length, but not entries to be missing at random intermediate points (see also S1 Supplementary Information, Section 1, for an illustration of how the established approach fails in this situation).

In this work, we propose and discuss three approaches of encoding missing data via fill-in values and augmenting the data. We integrate these into the BayesFlow workflow, and evaluate and compare their performance on three test problems. We find that an approach in which the data matrix is augmented with binary indicators of presence or absence of values performs the most robustly. Further, we demonstrate how this approach performs advantageously also in the simpler case of time series of different length, thereby showing how BayesFlow performance might be even improved in the absence of missing data.

## 1.1 Related work

There exist various possible missing data mechanisms, (not) missing (completely) at random (MCAR, MAR, NMAR, see [10]). Strategies to deal with missing data can be broadly divided into methods discarding entries with missing values, and approaches replacing missing values with imputed values. Imputation strategies include e.g. mean, maximum-likelihood, and multiple imputation, and deep learning approaches using e.g. LSTM networks [11] and autoencoders [12], see [13, 14] for a review. In the context of neural posterior estimation, [15] developed an integrated approach to simultaneously learn a model predicting parameters with failing

simulations, and impute missing data. However, the faithful reconstruction of missing data using (multiple or weighted) imputation approaches relies on the accuracy of the used imputation method (see also a discussion of the inadequacy of overly simple, e.g. linear, imputation methods in S1 Supplementary Information, Section 2). Moreover, when the uncertainty of imputed values is properly accounted for, by considering their distribution that depends on the available values and prior knowledge leveraged for imputation, parameter estimation from imputed data should yield the same effective posterior distribution as parameter estimation considering only the available data, provided the underlying missingness mechanism is captured accurately in the model (see S1 Supplementary Information, Section 3). Therefore, in such a Bayesian sense, there should be no conceptual advantage of using imputation over approaches discarding missing data, as long as they do not discard further data. Unfortunately, as many established inference methods cannot deal with missing (not-available, NA) entries, methods discarding missing entries often discard entire cases, such as listwise deletion [16], in which case they lose information. Our approach also discards missing values; however, importantly, it does not discard any non-missing data, and therefore neither loses information nor introduces a potential bias due to inadequacy of an imputation method. To the best of our knowledge, missing data discarding approaches for amortized inference have so far not been systematically studied.

## 2 Methods

### 2.1 Background

**2.1.1 Amortized simulation-based neural posterior estimation.** A mechanistic model induces a likelihood function $\theta \mapsto \pi(x|\theta)$ of measuring data $x \in \mathbb{R}^{n_x}$ given model parameters $\theta \in \mathbb{R}^{n_\theta}$. Parameter inference deals with the problem of estimating the unknown model parameters $\theta$, given experimentally observed data $x^{\mathrm{obs}} \in \mathbb{R}^{n_x}$. In a Bayesian setting, the likelihood is combined with prior information $\pi(\theta)$ on the parameters, giving by Bayes' Theorem the posterior distribution

$$\pi(\theta|x^{\mathrm{obs}}) \propto \pi(x^{\mathrm{obs}}|\theta)\pi(\theta). \tag{1}$$

There are two major challenges to working with (1): (i) In many applications, the mechanistic model is only available as a simulator, allowing to generate synthetic data $x \sim \pi(x|\theta)$, but not to evaluate the likelihood function $\pi(x^{\mathrm{obs}}|\theta)$ [17, 18]. (ii) Often, the same model needs to be fitted to different data sets $x^{\mathrm{obs},d}$, $d = 1, \ldots, n_d$, requiring the costly analysis of multiple posterior distributions.

BayesFlow approximates the posterior by a tractable distribution $\pi_\phi(\theta|x) \approx \pi(\theta|x)$ for any $(x, \theta) \sim \pi(x|\theta)\pi(\theta)$. The approximate posterior is parameterized in terms of a conditional invertible neural network (cINN) $f_\phi : \mathbb{R}^{n_\theta} \to \mathbb{R}^{n_\theta}$, $\theta \mapsto z$, conditioned on data $x$, which defines a normalizing flow [19] between the posterior over the parameters $\theta$ and a standard multivariate normal latent variable $z$,

$$\theta \sim \pi_\phi(\theta|x) \quad \Leftrightarrow \quad \theta = f_\phi^{-1}(z; x) \quad \text{with} \quad z \sim \pi_z(z) = \mathcal{N}_{n_\theta}(z|0, I_{n_\theta}).$$

The neural network parameters $\phi$ are trained to minimize the Kullback-Leibler (KL) divergence between posteriors over all possible data sets $x$,

$$\hat{\phi} = \arg\min_\phi \mathbb{E}_{x \sim \pi(x)}[\mathbb{KL}(\pi(\theta|x) \| \pi_\phi(\theta|x))] = \arg\max_\phi \mathbb{E}_{(\theta,x) \sim \pi(\theta,x)}[\log \pi_\phi(\theta|x)].$$

Via change of variable, it is $\pi_\phi(\theta|x) = \pi_z(f_\phi(\theta; x)) \cdot |\det J_{f_\phi}(\theta; x)|$, with straightforward calculation of the Jacobian $J_{f_\phi}(\theta; x) = \frac{\partial}{\partial \theta} f_\phi(\theta; x)$ due to the cINN architecture. In practice, a Monte-

Carlo approximation

$$\hat{\phi} \approx \arg\min_{\phi} \ \frac{1}{M}\sum_{m=1}^{M}\left(\frac{1}{2}\left\|f_\phi\big(\theta^{(m)};x^{(m)}\big)\right\|_2^2 - \log|\det J_{f_\phi}\big(\theta^{(m)};x^{(m)}\big)|\right)$$

of the expectation is employed, with samples $\{(\theta^{(m)}, x^{(m)}) \sim \pi(\theta, x)\}_{m=1}^{M}$.

**2.1.2 Invertible architecture.** The basic unit of the cINN in BayesFlow is the affine coupling layer [8, 20]. A layer comprises four internal functions $s_1, t_1, s_2, t_2$, each realized as fully connected neural networks with exponential linear units. In the forward direction, the input vector $u$ is split into two halves $u_1$ and $u_2$, which then undergo a sequence of operations to give the output $v = (v_1, v_2)$:

$$v_1 = u_1 \odot \exp(s_1(u_2)) + t_1(u_2), \quad v_2 = u_2 \odot \exp(s_2(v_1)) + t_2(v_1)$$

Here, $\odot$ denotes the Hadamard product and $\exp(\cdot)$ the elementwise exponential function. The non-linear mapping $u \mapsto v$ is bijective, with the inverse given by:

$$u_2 = (v_2 - t_2(v_1)) \odot \exp(-s_2(v_1)), \quad u_1 = (v_1 - t_1(u_2)) \odot \exp(-s_1(u_2))$$

In practice, multiple layers are stacked to craft an invertible chain with sufficient expressiveness. The input to the first layer are the parameters $\theta$ of interest, and the output of the final layer are the latent variables $z$. To condition the mapping on the data $x$, either $x$ or summary statistics $h = h_\psi(x)$ thereof are additionally fed as input into all internal networks. The chain of conditional layers yields the cINN $f_\phi$, which is trained to learn the normalizing flow from $\theta$ to $z$ using information from the data. The complete forward and inverse pass through the cINN can be written as $z = f_\phi(\theta; h)$ and $\theta = f_\phi^{-1}(z; h)$, respectively.

**2.1.3 Summary networks.** Instead of feeding the raw data directly into the cINN, summary statistics $h = h_\psi(x)$ can be employed. This has two advantages [8]: First, tailored dimension reduction methods can adequately summarize redundant data and account for symmetries. Second, this enables the method to work with varying data set sizes $x \in \mathbb{R}^{n_x}$ with random $n_x \in \mathbb{N}$, by transforming them into fixed-size representations. For example, an LSTM [21] can handle time series $x$ of different length.

To avoid manual crafting, [8] propose to learn maximally informative statistics from the data, by training the summary network parameters $\psi$ jointly with the invertible network parameters, giving the joint objective

$$\hat{\phi}, \hat{\psi} \approx \arg\min_{\phi,\psi} \frac{1}{M}\sum_{m=1}^{M}\left(\frac{1}{2}\left\|f_\phi\big(\theta^{(m)};h_\psi(x^{(m)})\big)\right\|_2^2 - \log|\det J_{f_\phi}\big(\theta^{(m)};h_\psi(x^{(m)})\big)|\right) \quad (2)$$

It can be shown that, provided sufficient training and expressiveness of $f_\phi$ and $h_\psi$, $\pi_{\hat{\phi}}(\theta|h_{\hat{\psi}}(x)) = \pi(\theta|x)$ almost surely, i.e. the learned posterior perfectly approximates the actual one.

The upfront *training phase* can be expensive, as it might require many model simulations. Once the approximate posterior $\pi_{\hat{\phi}}(\theta|h_{\hat{\psi}}(x))$ has been trained, for the *inference phase*, observed data $x^{\text{obs}}$ are passed through the summary network, $h^{\text{obs}} = h_\psi(x^{\text{obs}})$. Then, latent variables $z^{(l)} \sim \mathcal{N}_{n_\theta}(0, I_{n_\theta}), l = 1, \ldots, L$ are sampled and transformed to samples $\theta^{(l)} = f_{\hat{\phi}}^{-1}(z^{(l)}; h^{\text{obs}}) \sim \pi_{\hat{\phi}}(\theta|h_{\hat{\psi}}(x))$ from the target distribution, by passing them through the cINN in inverse direction.

BayesFlow tackles both challenges (i) and (ii) above: It does not require likelihood evaluations and can thus be applied to any simulator. Further, it gives posterior approximations for

any possible parameters and data. The inference phase is relatively cheap, as it does not require simulations of the mechanistic model, allowing to amortize the training phase when applied to different data sets $x^{\mathrm{obs}}$.

## 2.2 Encoding missing data

In many applications, not all data sets are complete. A prime example are clinical data, in which patients might have missed an appointment or dropped out of a study. In the case of incomplete data, the vector $x^{\mathrm{obs}}$ is not available completely, but some of its entries are missing, giving observed data $x^{\mathrm{obs}}_{\mathrm{avai}} \in (\mathbb{R} \cup \{\mathrm{NA}\})^{n_x}$. Explicitly, defining the binary availability mask $\tau^{\mathrm{obs}} \in \{0, 1\}^{n_x}$, where 0 indicates absence and 1 presence of a data point, we consider $x^{\mathrm{obs}}_{\mathrm{avai}} = x^{\mathrm{obs}} \odot \tau^{\mathrm{obs}} + \mathrm{NA} \cdot (1 - \tau^{\mathrm{obs}})$, where the convention $\mathrm{NA} \cdot 1 = \mathrm{NA}$ and $\mathrm{NA} \cdot 0 = 0$ is used. There can be many causes and patterns of missingness, e.g. completely at random or dependent on the parameters. We are agnostic of the exact underlying mechanisms, and interested in the posterior $\pi(\theta|x^{\mathrm{obs}}_{\mathrm{avai}})$ conditioned only on the available data. A different problem would be posed by the posterior $\pi(\theta|x^{\mathrm{obs}})$ with $x^{\mathrm{obs}}$ the full data, which would however require a (faithful, multiple) reconstruction.

Missing values ("not available", "not a number") cannot be handled by neural networks, as they result in failing cost function and gradient evaluations. Simply dropping them from the data vector is no solution in the context of amortized inference, as the information about which data were present would be lost. Further, the BayesFlow cINN $f_\phi$ requires inputs of fixed size across samples. A summary network $h_\psi$ that permits inputs of variable size and transforms them into fixed-size representations constitutes one solution. This renders the method applicable to e.g. time series of different length, which can be mapped by an LSTM to a fixed-size latent state. However, this does not extend to random intermediate missing entries.

Here, we propose three ways of handling data sets $x_{\mathrm{avai}}$ with missing entries, to enable inference on the posterior $\pi(\theta|x_{\mathrm{avai}})$. For simplicity, assume that the mechanistic model produces data sets of fixed size $n_x$. This is in practice no restriction, as we can embed data sets of different size into one of maximum dimension $n_x^{\mathrm{max}}$, considering entries $n_x < i \le n_x^{\mathrm{max}}$ as missing too. We propose the following ways of encoding missing data into a vector $x_{\mathrm{aug}}$ such that the neural network can learn to detect and ignore them (see Fig 1 for an illustration of the resulting full workflow and examples):

- E1 ("Insert $c$"): Insert a constant value $c \in \mathbb{R}$ in place of missing data points. That is, set $x_{\mathrm{aug}} := \iota_c(x_{\mathrm{avai}}) := x_{\mathrm{avai}} \odot \tau + c \cdot (1 - \tau)$, i.e. $\iota(\mathrm{NA}) = c$, and $\iota(x) = x$ for $x \in \mathbb{R}$.

- E2 ("Augment by 0/1"): As in E1, insert a constant value $c \in \mathbb{R}$ in place of missing data points. In addition, augment the data $x_{\mathrm{avai}}$ by the availability mask $\tau \in \{0, 1\}^{n_x}$ to a combined matrix $x_{\mathrm{aug}} := [\iota_c(x_{\mathrm{avai}}), \tau]$.

- E3 ("Time labels"): Restrict the data $x_{\mathrm{avai}}$ to available entries, yielding a reduced vector $(x_{\mathrm{avai}})_\tau \in \mathbb{R}^{\sum_i \tau_i}$. Augment this vector by a mask of time point indices $l \in \mathbb{R}^{\sum_i \tau_i}$, or alternative positional encodings, giving a combined data matrix $x_{\mathrm{aug}} := [(x_{\mathrm{avai}})_\tau, l]$.

## 2.3 Training algorithm with missing data

In order to facilitate recognizing the encoding of incomplete experimentally observed data during the inference phase, we train BayesFlow on data sets containing missing entries. Therefore, given complete simulated data $x^{(m)} \sim \pi(x|\theta)$, we explicitly simulate an availability pattern

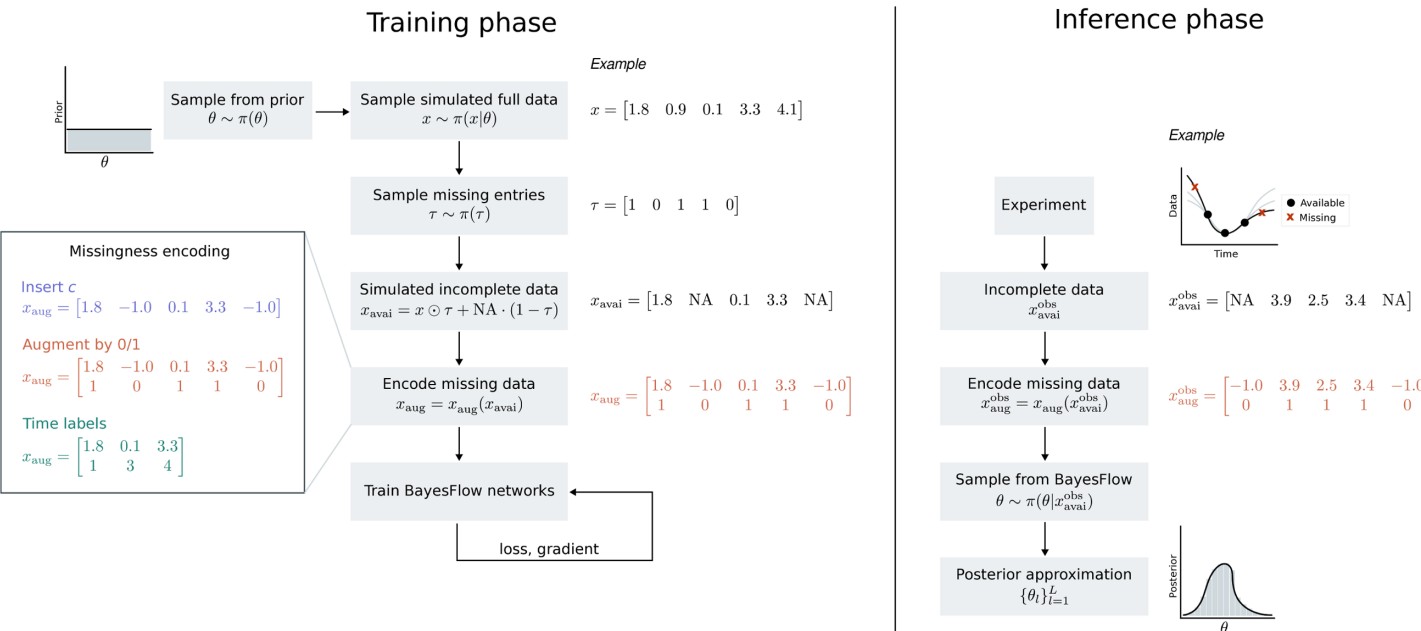

**Fig 1. Illustration of the workflow combining BayesFlow with missing data encoding.** Upfront training phase (left): Parameters $\theta \sim \pi(\theta)$ are sampled from the prior to simulate complete data sets $x_{1:N}$. Then, missing entries are randomly selected and encoded according to one of the three approaches "Insert $c$" (here $c = -1$), "Augment by 0/1" (here $c = -1$), and "Time labels". The BayesFlow network is trained on such data sets with missing values using an online learning algorithm. Amortized inference (right): Experimentally observed incomplete data $x_{\text{avai}}^{\text{obs}}$ are processed using the preferred encoding approach (here "Augment by 0/1", $c = -1$) and passed through the pre-trained BayesFlow network in its inverse direction. This leads to representative samples from the posterior conditioned on the available data $\pi(\theta | x_{\text{avai}}^{\text{obs}})$. The upfront training amortizes over inference on arbitrarily many incomplete data sets.

$\tau^{(m)} \sim \pi(\tau) \in \{0, 1\}^{n_x}$ in order to generate artificially incomplete data $x_{\text{avai}}^{(m)} = x^{(m)} \odot \tau^{(m)} + \text{NA} \cdot (1 - \tau^{(m)})$. These are then passed through one of the missingness encoders, giving augmented data $x_{\text{aug}}^{(m)}$, which are fed into summary and invertible network. The distribution of available entries $\pi(\tau)$ should incorporate any prior knowledge on missingness patterns, in order to train the network on realistic scenarios. At its simplest, we consider uniformly missing entries, i.e. with $n_{\varnothing}^{\max} \leq n_x$ the maximum number of missing entries, we sample $n_{\varnothing}^{(m)} \sim \mathcal{U}(0, n_{\varnothing}^{\max})$, and then sample without replacement $n_{\varnothing}^{(m)}$ indices out of $[1, \ldots, n_x]$.

In order to vectorize the propagation of samples through summary and invertible network, as well as gradient calculation via backpropagation, all samples within a batch must be of the same size. Thus, for approach E3 above, a single number $n_{\varnothing}$ of missing data points needs to be sampled per batch, like in the original BayesFlow implementation when considering time series of different length. The exact distribution of the $n_{\varnothing}$ missing entries over the data set can still be individual-specific. Meanwhile, in E1+2 the augmented data dimension is fixed and independent of the number of missing entries, such that the number of missing data points can be sampled for each sample individually. Sampling the number of missing data points on individual instead of batch level reduces the variance of the cost function approximation across batches (i.e. iterations), which can be hoped to improve the stability of gradient descent when training the network parameters.

The entire algorithm for training and inference using BayesFlow with missing data is presented in Algorithm 1. We use an online learning approach similarly to [8], meaning that a batch of new data is simulated in each iteration. This is reasonable, since model simulations in our test problems are relatively fast, and effective to prevent overfitting. If simulations are

more expensive, one should consider approaches where simulated data are reused to speed up training, possibly combined with regularization techniques.

**Algorithm 1** Amortized Bayesian inference for incomplete data using BayesFlow

1: **Training phase** (*online learning using data sets with artificially induced missing values*):
2: *Input*: Prior $\pi(\theta)$, simulator $\pi(x|\theta)$, missingness pattern $\pi(\tau)$, batch size $M$.
3: **repeat**
4:   If using encoding E3, sample number of missing entries $n_\varnothing \sim \pi(n_\varnothing | n_x)$ (or the entire availability pattern $\tau \sim \pi(\tau)$).
5:   **for** $m = 1, \ldots, M$ **do**
6:     Sample model parameters from the prior: $\theta^{(m)} \sim \pi(\theta)$.
7:     Generate a synthetic complete data set: $x^{(m)} \sim \pi(x|\theta^{(m)})$.
8:     If using encoding E1 or E2, sample availability pattern $\tau^{(m)} \sim \pi(\tau)$.
9:     If using encoding E3, sample availability pattern $\tau^{(m)} \sim \pi(\tau|n_\varnothing)$ (or set $\tau^{(m)} = \tau$).
10:      Mask missing entries: $x_{\mathrm{avai}}^{(m)} = x^{(m)} \odot \tau^{(m)} + \mathrm{NA} \cdot (1 - \tau^{(m)})$.
11:      Encode $x_{\mathrm{avai}}^{(m)}$ via E1–3, yielding augmented data $x_{\mathrm{aug}}^{(m)}$.
12:      Pass the augmented data through the summary network: $h^{(m)} = h_\psi(x_{\mathrm{aug}}^{(m)})$.
13:      Pass ($\theta^{(m)}$, $h^{(m)}$) through the inference network in forward direction: $z^{(m)} = f_\phi(\theta^{(m)}; h^{(m)})$.
14:   **end for**
15:   Compute the loss $\mathcal{L}(\phi, \psi)$ according to (2) from the training batch $\{(\theta^{(m)}, h^{(m)}, z^{(m)})\}_{m=1}^M$.
16:   Update neural network parameters $\phi$, $\psi$ via backpropagation.
17: **until** convergence to $\hat{\phi}, \hat{\psi}$
18: Return $\hat{\phi}, \hat{\psi}$.
19:
20: **Inference phase** (*given an incomplete observed data set*):
21: *Input*: Observed incomplete data $x_{\mathrm{avai}}^{\mathrm{obs}}$, number of posterior samples $L$.
22: Encode $x_{\mathrm{avai}}^{\mathrm{obs}}$ via E1–3, yielding augmented data $x_{\mathrm{aug}}^{\mathrm{obs}}$.
23: Pass the augmented data through the summary network, yielding $h^{\mathrm{obs}} = h_{\hat{\psi}}(x_{\mathrm{aug}}^{\mathrm{obs}})$.
24: **for** $l = 1, \ldots, L$ **do**
25:   Sample a latent variable instance: $z^{(l)} \sim \mathcal{N}_{n_\theta}(z|0, I_{n_\theta})$.
26:   Pass ($z^{(l)}$, $h^{\mathrm{obs}}$) through the inference network in inverse direction, yielding $\theta^{(l)} = f_{\hat{\phi}}^{-1}(z^{(l)}; h^{\mathrm{obs}})$.
27: **end for**
28: Return $\{\theta^{(l)}\}_{l=1}^L$ as a sample from $\pi(\theta|x_{\mathrm{avai}}^{\mathrm{obs}})$.

## 2.4 Implementation

Unless otherwise stated, we used the default settings of BayesFlow (version 0.0.0b1). For all problems, we ran the training phase for 300 epochs, each consisting of 1000 iterations, with one iteration denoting one batch of samples, over which the loss was calculated and backpropagation performed. The batch size was 64 or 128, depending on the complexity of model simulations. As summary network, we used an LSTM with the number of hidden units being a power of two close to the data dimension. Unless otherwise specified, we used the actual time points as positional encoding in the "Time labels" method. The analyses were performed on a single CPU (AMD EPYC 7443 2.85 GHz) with 48 cores and 1 TB RAM. The full code

underlying this study can be found at https://github.com/emune-dev/Data-missingness-paper, a snapshot of code and data is available on Zenodo at https://doi.org/10.5281/zenodo.7515458.

# 3 Results

We evaluated and compared the performance of the proposed missing data encodings E1–3 on five test problems—a simple conversion reaction model, a sinusoidal model, the FitzHugh-Nagumo (FHN) neuron model, and ODE and SSA versions of the SIR epidemiological model. Details on all test problems can be found in S1 Supplementary Information, Section 4. In S1 Supplementary Information, Section 5, we provide further analyses e.g. on convergence for all models, beyond the main results shown here in the main manuscript. When comparing results to the "true posterior", we imply the distribution $\pi(\theta|x_{\text{avai}}^{\text{obs}})$ given only actually observed data, ignoring missing entries, as we assume ignorable missingness in the considered problems. This reference was computed analytically for the conversion reaction, sinusoidal and SIR ODE model, approximated numerically using Markov Chain Monte Carlo for the FHN model, and using approximate Bayesian computation for the SIR SSA model.

## 3.1 Conversion reaction model

We considered an ordinary differential equation (ODE) model of a simple conversion reaction $A \rightleftharpoons B$, a common building block in many biochemical systems and a widely used test problem (see e.g. [22, 23]). We assumed additive normally distributed measurement noise and that up to $n_{\varnothing}^{\max} = 2$ of the overall $n_x = 3$ observations are missing. The aim is to infer the posterior of the two log-scale rate parameters $k_1, k_2$.

**3.1.1 All approaches perform well on simple test problem.**   To assess the performance of different encodings, we trained for each a 5-layer cINN with an LSTM with 8 hidden units as summary network. For "Insert $c$" and "Augment by 0/1", we used a fill-in value of $c = -1$, as the model only allows positive trajectories.

Our assessment of the results revealed that for this simple problem, all three encodings led to accurate posterior approximations, even though the true posterior can be clearly non-Gaussian (Fig 2, Data set 1). In addition, in the special case that both values at $t_2$ and $t_3$ are missing and only the initial time point $t_1$ is available, we are dealing with a completely uninformative data set, and all three approaches correctly returned the Gaussian prior distribution (Fig 2, Data set 2). This indicates that BayesFlow can conceptually comprehend each of the encodings, and focus on the available data points while ignoring masked missing entries.

**3.1.2 Binary indicator augmentation increases robustness.**   In some applications no suitable dummy imputation value might be known a-priori, e.g. when the model is flexible enough to simulate unbounded trajectories, when the prior range encompasses experimentally implausible regimes, or when noise levels can be large. In this case, the network might have difficulties to distinguish between the imputed and the really occurring value, and the approaches "Augment by 0/1" and "Insert $c$" may have difficulties in distinguishing between signal and dummy imputation values representing missing data. To assess this, we tested them on the conversion reaction model with an ambiguous dummy value of $c = 0.5$. We observed that the approach "Insert 0.5" misinterprets measured values in a neighborhood of 0.5 as missing data. In contrast, the approach "Augment 0/1", additionally employing binary indicator augmentation, is capable of deciding correctly whether a value around 0.5 represents signal or missing value (Fig 3).

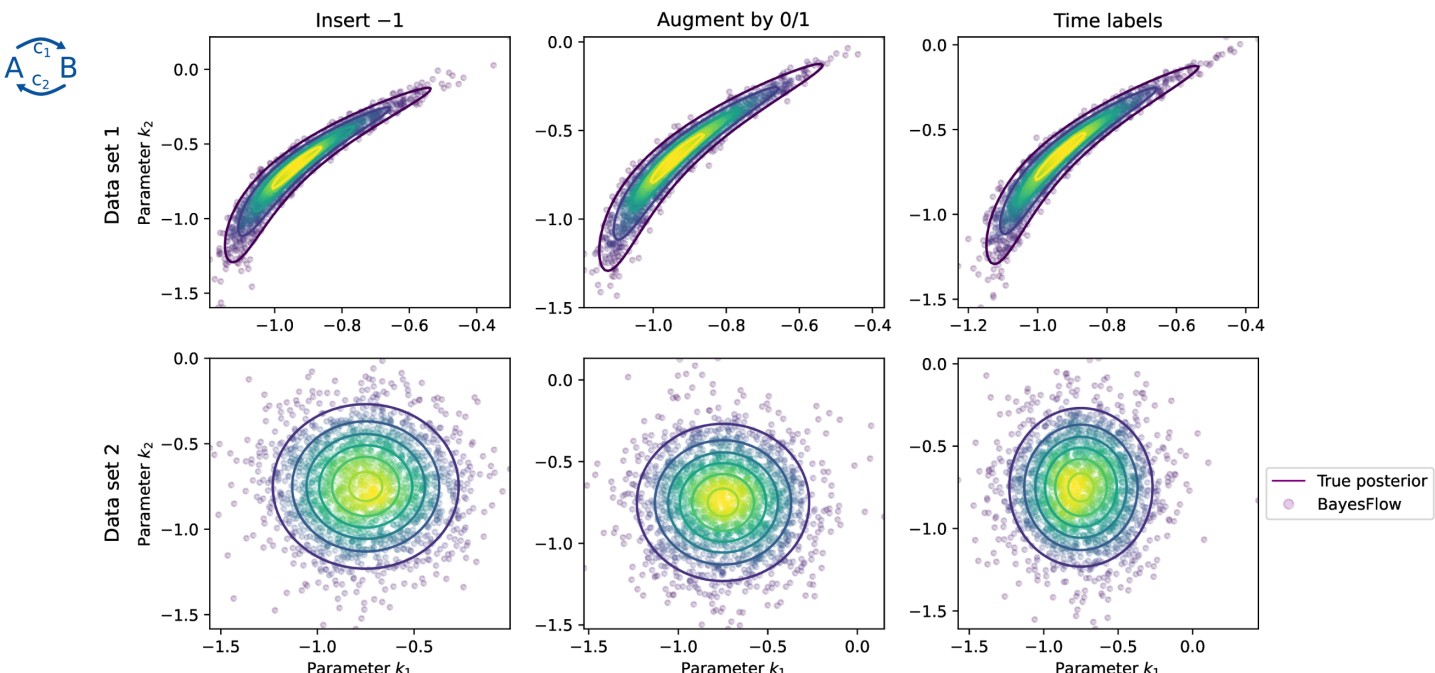

**Fig 2.** *Posterior approximations for the conversion reaction model* with $n_x = 3$ observations. Two test data sets at parameters $[-0.98, -0.66]$ (Data set 1, top, $n_\varnothing = 1$) and $[-0.71, -0.54]$ (Data set 2, bottom, $n_\varnothing = 2$) are shown. In Data set 2, no informative data are available, such that the posterior must equal the prior. All three encodings yield near-perfect posterior approximations for this simple problem.

## 3.2 Oscillatory models

As a class of more complex test problems, we studied two oscillatory models. Oscillations play an important role in many biochemical systems, e.g. in the context of metabolism [24] and cell cycle [25]. From a mathematical perspective, models producing oscillatory data are in general hard to fit, as the landscape of the cost function can be highly irregular with multiple local minima [26]. We first considered a simple sinusoidal model given by the function $\sin(2\pi a \cdot t) + b$, in which we aim to infer frequency $a$ and offset $b$.

**3.2.1 Improved performance on variable data set size as a special case of missing data.** The original BayesFlow implementation [8] can already deal with time series models producing data sets of different length, by preprocessing the data with a suitable LSTM summary network, which reads in a given time series sequentially and summarizes it into a fixed-size representation, before feeding it into the cINN. Our missing data encoding also naturally covers this, as time series of different length can always be interpreted as data in which the last $n_x^{\max} - n_x$ time steps are missing.

To study this, we assumed the number of observations in the sinusoidal model to vary uniformly between $n_x^{\min} = 2$ and $n_x^{\max} = 41$. For both the original BayesFlow method and the approach "Augment by 0/1 ($c = -5$)", we trained a 5-layer cINN jointly with an LSTM with 128 hidden units as summary network. Comparing the losses over 300 epochs (Fig 4A), we observed that for "Augment by 0/1" the loss converged faster, more smoothly and towards a slightly lower final value, which corresponds to a better approximation of posterior distributions.

This can be explained by the fact that in the original BayesFlow method the number of available data points is sampled only once per batch, to enable vectorized operations. This

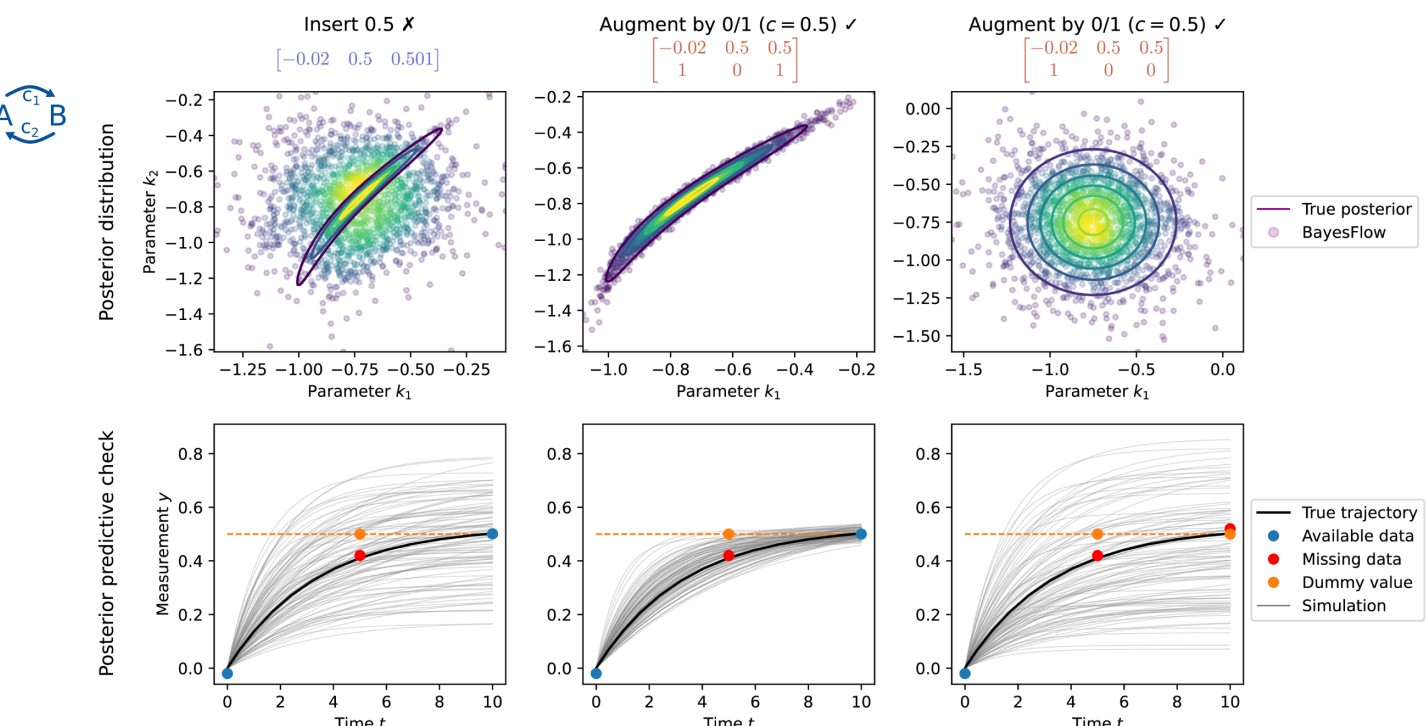

**Fig 3. *Increased robustness through binary indicator augmentation in case of ambiguous dummy values $c = 0.5$ for the conversion reaction model with $n_x = 3$ observations.*** Left: The approach "Insert 0.5" sees a data set in which only the second observation is missing. However, the network misinterprets the signal 0.501 as another missing value. Hence, the estimated posterior is wrong, and the third available data point is not fitted by the re-simulated trajectories. Middle: The approach "Augment by 0/1" is able to correctly identify the value 0.5 in the second entry as a missing value and in the third entry as a signal. Consequently, the estimated posterior is correct, and the re-simulated trajectories fit the third data point, but not the second one. Right: With changed binary indicator, the approach "Augment by 0/1" correctly interprets the value 0.5 in the second and third entry as missing, despite 0.5 being a plausible data value for the third entry.

results in a loss function estimate with substantially higher variance (although it remains unbiased). Its severe fluctuation over iterations (Fig 4B) may then affect the convergence of the stochastic gradient descent algorithm (Fig 4A). Contrarily, the approach "Augment by 0/1" yields augmented data sets of fixed size, thus it is possible to sample the number of available data

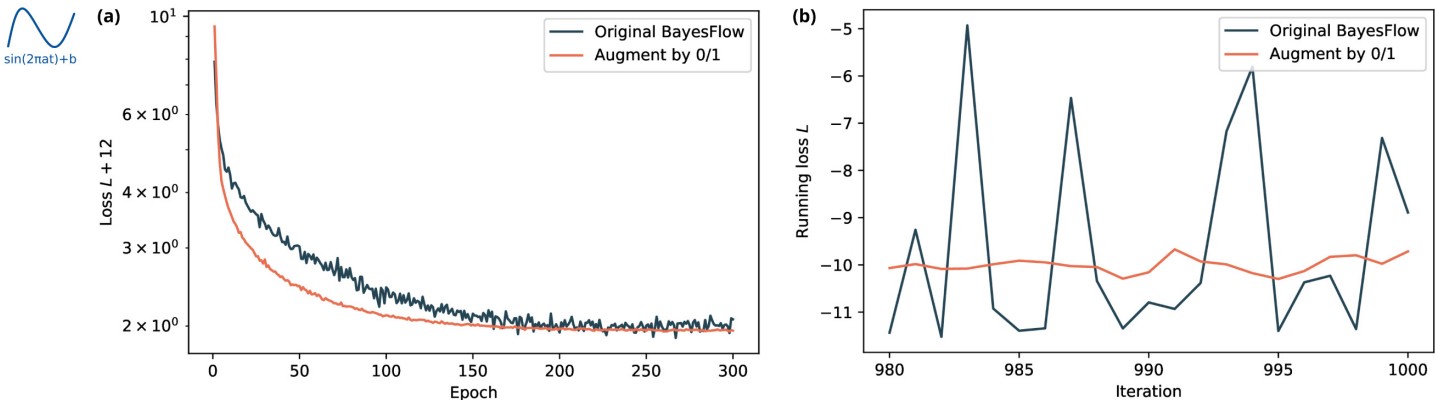

**Fig 4. Comparison of loss behavior for the sinusoidal model with variable data set length.** (a) Epoch-averaged loss over all 300 training epochs. (b) Loss in the last 20 iterations of the final epoch. Our missing data handling approach based on binary indicator augmentation achieves superior convergence to the original BayesFlow method, both globally (a) and on the level of individual iterations (b).

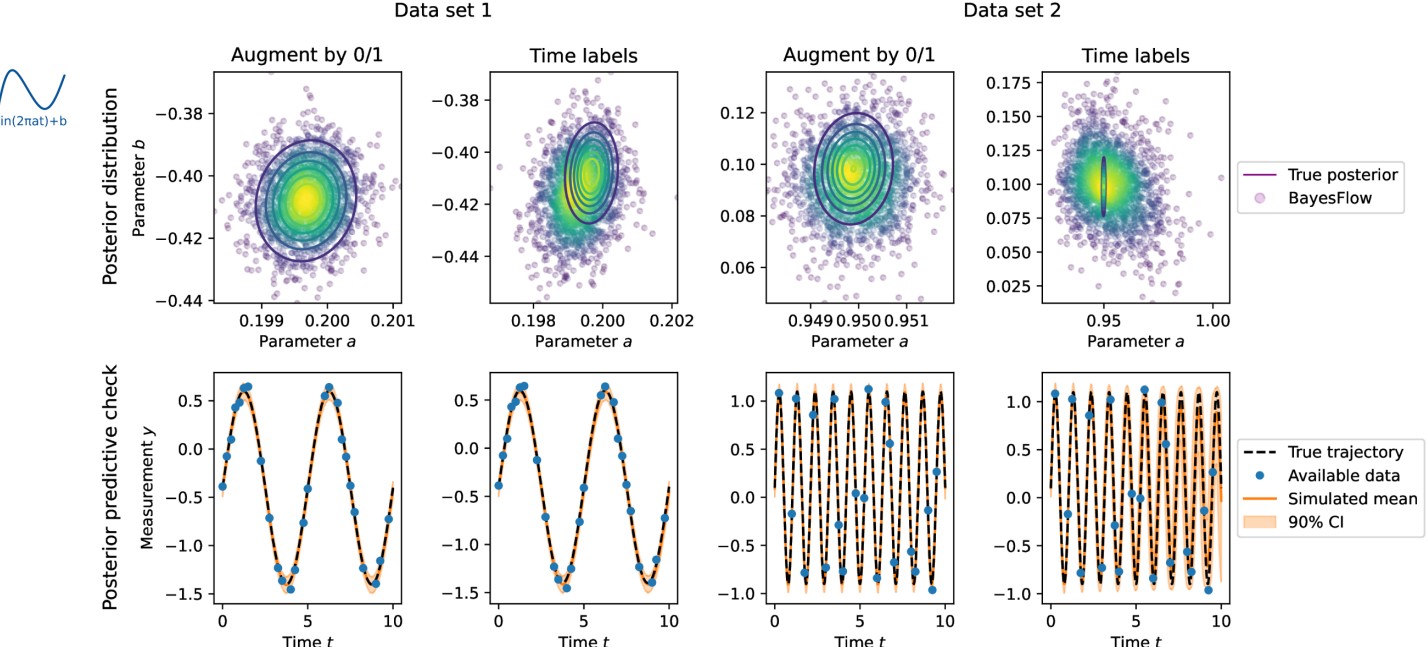

**Fig 5. Results for the sinusoidal model with uniformly sampled missing time steps.** Top: Posterior distributions. Bottom: Posterior predictive checks. Two data sets at parameters [0.2, −0.4] (Data set 1, left, $n_\varnothing = 15$) and [0.95, 0.1] (Data set 2, right, $n_\varnothing = 20$) are shown.

points, or here rather the time series length, on the level of individual data sets and thereby calculate loss estimates with reduced variance.

**3.2.2 "Time labels" encoding performs poorly on sinusoidal model.** Assuming uniform missingness with a maximum number of missing time steps of $n_\varnothing^{\max} = 21$ in the sinusoidal model, we compared the performance of the two encodings "Augment by 0/1" and "Time labels". We trained a 5-layer cINN with an LSTM with 128 hidden units as summary network. A dummy value of $c = -5$ was employed for the former encoding.

Across test data sets generated for different ground truth parameters and exhibiting differently many missing values, we consistently observed the approach "Augment by 0/1" to approximate the true posterior better than the approach "Time labels" (Fig 5). The latter approach suffers, similarly to the original BayesFlow algorithm in the previous section, from sampling the number of missing values only once per iteration and not per individual simulation. Consequently, its cost function exhibited substantially more fluctuations compared to "Augment by 0/1" (see S1 Supplementary Information, Fig K). Similarly increased fluctuations could be observed when sampling the number of missing observations on batch level for the "Augment by 0/1" approach. However, in addition "Time labels" converged to an altogether worse cost function value. A similar value was obtained by "Augment by 0/1" already after about 25 epochs. Comparing its posterior approximation at this early training stage with the final posterior approximation obtained by "Time labels", we saw qualitatively similar results (S1 Supplementary Information, Fig L). Also using an alternative attention-based transformer summary network, designed to work with positional encodings, did not improve the performance of "Time labels" (S1 Supplementary Information, Fig P and Fig Q). While further analyses would be needed, this could indicate that the BayesFlow network misinterpreted the time labels and could thus not converge to the true distribution.

**3.2.3 Similar behavior on FitzHugh-Nagumo model.** To check our findings on the sinusoidal model, we studied another oscillatory model, the FitzHugh-Nagumo (FHN) model, an ODE model with three parameters describing excitable systems [26]. Assuming a variable data set size between $n_x^{\min} = 2$ and $n_x^{\max} = 21$, we compared the original BayesFlow method with the approach "Augment by 0/1 ($c = -5$)" by training a 5-layer cINN with an LSTM with 64 hidden units as summary network. Although the original BayesFlow method again showed a more fluctuating loss than "Augment by 0/1", this time both converged comparably well (S1 Supplementary Information, Fig R).

Assuming that data are missing uniformly at up to $n_\varnothing^{\max} = 11$ of the overall $n_x = 21$ time steps, we compared "Augment by 0/1" and "Time labels" by training a 5-layer cINN each with an LSTM with 64 hidden units as summary network. Qualitatively similar results were achieved compared to the sinusoidal model, i.e. worse loss function convergence and worse posterior approximations by "Time labels" than by "Augment by 0/1", albeit less pronounced (S1 Supplementary Information, Fig T to Fig W).

In summary, the "Time labels" encoding appears to have problems with oscillatory models, whereas "Augment by 0/1" performed robustly. The underlying reason remains to be investigated.

## 3.3 SIR epidemiological models

Compartmental models have been widely used to describe the course of the COVID-19 pandemic [27, 28]. However, infectious disease data are almost always subject to missing values. Therefore, we next studied two SIR-type models, the first one modeling the involved compartments as an ODE [29], and the second one as a discrete Markov process via the stochastic simulation algorithm (SSA) [8]. In both cases, we aimed to estimate transmission and recovery rate parameters. We assumed uniform missingness of at most $n_\varnothing^{\max} = 15$ of the overall $n_x = 21$ time steps, and trained a 5-layer cINN with an LSTM with 128 hidden units as summary network.

For the encoding "Augment by 0/1" with a dummy value of $c = -1$, we obtained precise posterior approximations and data fits. This shows that this approach can deal with both more complex models and a high degree of missingness (Fig 6 for the ODE model; and S1 Supplementary Information, Section 5.6 for the SSA model). In particular, this renders possible simulation-based inference for stochastic models with missing data in an amortized fashion. See further S1 Supplementary Information, Sections 5.5 and 5.6 for "Insert −1", which performed comparably, and "Time labels", which converged slightly worse on these problems.

## 3.4 Parameter-dependent missingness can be captured

In the above test problems, missing entries were always sampled uniformly and independently of the parameters. This may however not be the case in real-world applications, e.g. the frequency of case numbers being reported in an epidemic may depend heavily on the disease severity.

Therefore, as a prototype for dynamical models exhibiting parameter-dependent missingness, we modified the conversion reaction model with up to $n_x = 11$ observations by adding a missingness parameter $p \sim \mathcal{U}(0, 0.8)$ that determines the portion of missing values via $n_\varnothing = \lfloor p \cdot n_x \rfloor$. Then, $n_\varnothing$ values of the $n_x$ are uniformly sampled to be missing (Fig 7). For simplicity, we fixed the rate parameter $k_2$. Employing the encoding "Augment by 0/1" with a dummy value of $c = -1$, we trained a 5-layer cINN with an LSTM with 32 hidden units as summary network to jointly estimate $k_1$ and $p$.

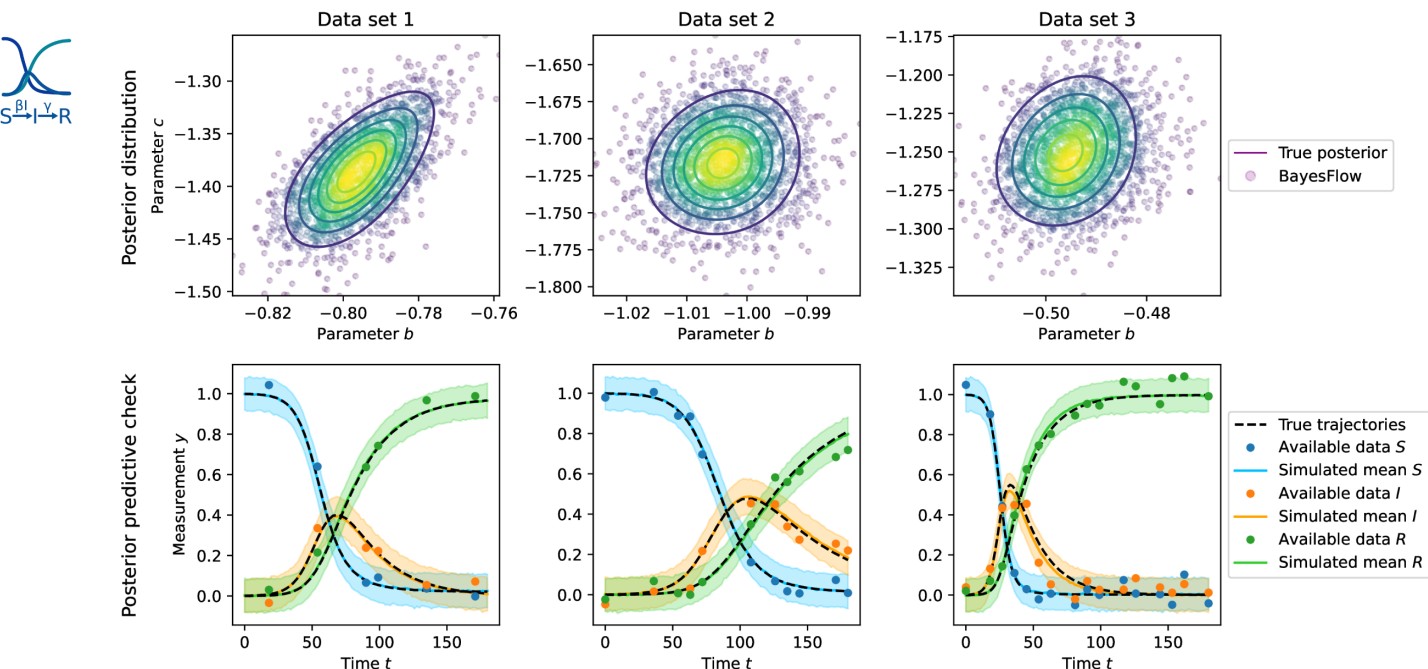

**Fig 6. Results for the SIR ODE model.** Top: Posterior distributions. Bottom: Posterior predictive checks displaying the means of noise-corrupted simulations and their centered 90% credible intervals. Three data sets at ground truth parameters $[-0.8, -1.4]$ (Data set 1, left, $n_\varnothing = 15$), $[-1.0, -1.7]$ (Data set 2, middle, $n_\varnothing = 10$) and $[-0.5, -1.3]$ (Data set 3, right, $n_\varnothing = 5$) are shown.

Across different test data, we observed that our method not only correctly identified the interval of parameter values for $p$ that leads to the observed number of missing values, but also captured the independence between the dynamics parameter $k_1$ and the missingness parameter $p$ (Fig 7). This shows that our approach is conceptually able to unravel parameter-missingness dependencies.

## 4 Discussion

Motivated by the fact that missing data are ubiquitous in experimental studies, in this work we presented approaches that allow to adequately handle them while performing inference over

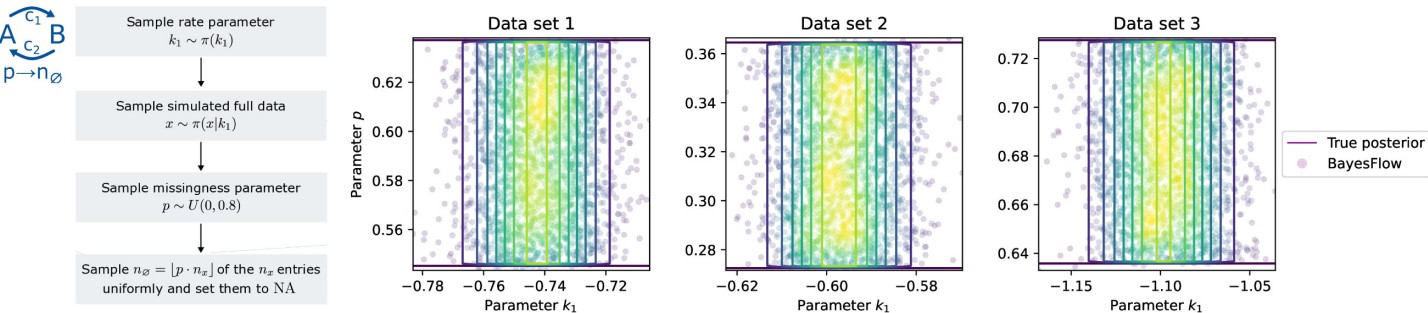

**Fig 7. _Parameter-dependent missingness_ for a modified conversion reaction model.** Left: Visualization of the data generation process. Right: Posterior approximations using the encoding "Augment by 0/1" for three data sets at ground truth parameters $[-0.75, 0.6]$ (Data set 1), $[-0.6, 0.3]$ (Data set 2) and $[-1.1, 0.7]$ (Data set 3).

many observed data sets simultaneously. We achieved this by encoding the missing entries by fill-in values ("Insert $c$") and augmenting the data by a binary mask indicating absence or presence ("Augment by 0/1"), or a mask identifying the available data points globally ("Time labels"), and using the established BayesFlow framework.

In particular, we found the approach "Augment by 0/1" to perform robustly across different problems. Unlike "Insert $c$", it provides a binary indicator which can be easily interpreted by the neural network. This renders the approach particularly useful in case of ambiguous fill-in values. However, this comes at the cost of increased effective data set size. Thus, in cases a clear fill-in value can be found, "Insert $c$" may suffice and perform more efficiently. On the considered examples, we observed no substantial run time differences between the approaches.

In the original BayesFlow implementation, time series lengths were sampled only once per batch. This was in order to facilitate vectorized operations when propagating through the network. We showed that sampling the missingness pattern per individual, rather than once per batch, improves convergence, as it leads to a more stable Monte-Carlo approximation of the cost function. In particular, this renders "Augment by 0/1" a superior alternative to the "Time labels" approach, and also to the original BayesFlow implementation in the simplified case of time series of different lengths. The approaches are broadly applicable to various missingness scenarios. In particular, they allow also to handle irregular time series data, via projecting onto a high grid resolution with missing entries.

While we obtained some first promising results on how to combine amortized inference with missing data, several questions remain open:

It remains to be studied how the suggested missingness encodings perform comparatively on more challenging application problems. Such problems would allow to realistically benchmark the approaches, e.g. on the trade-off of augmented data set size and accuracy, and to further evaluate the ability to unravel parameter-dependent missingness patterns.

Instead of sampling the missingness pattern individually per model simulation, one could generate multiple data sets with missing entries from a single full simulation. Especially for expensive mechanistic models, this could be useful. Moreover, this could be easily combined with an offline training approach, where, in contrast to the online training approach employed throughout this study, simulated data using the mechanistic model are generated only once before the analysis. In such an offline approach, missing entries for each full simulation (and more generally parts of the model simulation that are computationally inexpensive) could be generated anew in each generation to avoid overfitting while keeping the computational advantages of offline training. However, the resulting trade-off of simulation cost and accuracy or convergence needs to be investigated.

Further, attention-based networks such as transformers are designed to work with positional encodings, such as the here-used time labels. However, on oscillatory data we observed inferior performance, which remains to be studied in detail.

Moreover, while we here provided an integration into the BayesFlow methodology, the encodings may also prove useful for other amortized inference approaches, such as [30], which however remains to be studied.

Further, as we discussed briefly, an alternative to ignoring missing data is to impute faithful values. An investigation of the applicability of such approaches and a comparison to the here presented missingness encodings in terms of the obtained posterior approximation would be of interest.

Last, in this study we have considered entries missing completely at random (MCAR), or with the portion of missing entries dependent on latent parameters. Inspection of the derivation of our approach in Section 2.3 however reveals that no assumptions on the missingness distribution $\pi(\tau)$ need to be made, as it is completely simulation-based. In particular, this

conceptually allows the consideration of general data-dependent missingness patterns $\pi(\tau|x)$ (MNAR). However, the practical feasibility of this remains to be tested. As common in missing data problems, an application to MNAR can be challenging, as it requires an accurate specification of the distribution of missing values given observable data. In general for neural posterior estimation, adequacy of the learned posterior distribution across the domain of possible observed data and parameters, as well as adequacy of the employed model to describe the observed data must be carefully assessed. For this, there exist established systematic approaches, whose application in the context of missing data would however remain to be checked [8, 31].

In conclusion, in this work we presented and compared approaches to handle missing data in the amortized simulation-based neural posterior estimation framework BayesFlow. We believe that this will substantially improve its applicability on a wide range of problems.

## Supporting information

**S1 Supplementary Information. Additional analyses and information supplementing this manuscript.**
(PDF)

## Acknowledgments

We thank Ullrich Köthe, Stefan Radev, and Marvin Schmitt for fruitful discussions, and Dilan Pathirana for assistance with computational resources.

## Author Contributions

**Conceptualization:** Jan Hasenauer, Yannik Schälte.

**Formal analysis:** Zijian Wang, Yannik Schälte.

**Methodology:** Zijian Wang.

**Project administration:** Jan Hasenauer.

**Supervision:** Jan Hasenauer, Yannik Schälte.

**Validation:** Zijian Wang.

**Visualization:** Zijian Wang.

**Writing – original draft:** Zijian Wang, Jan Hasenauer, Yannik Schälte.

**Writing – review & editing:** Zijian Wang, Jan Hasenauer, Yannik Schälte.

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
