## [Decision Letter · Decision Letter 0]

13 Mar 2024

Dear Mr. Schälte,

Thank you very much for submitting your manuscript "Missing data in amortized simulation-based neural posterior estimation" for consideration at PLOS Computational Biology. Your manuscript was reviewed by members of the editorial board and an independent reviewer, who appreciated the attention to an important topic. Based on the reviews, we are likely to accept this manuscript for publication, providing that you modify the manuscript according to the review recommendations.

Sincerely,

James R. Faeder

Academic Editor

PLOS Computational Biology

Pedro Mendes

Section Editor

PLOS Computational Biology

Reviewer's Responses to Questions

**Comments to the Authors:**

Reviewer #1: This paper extends neural posterior estimation (NPE), in particular the BayesFlow method, to incorporate missing data. It compares 3 natural methods for doing this and, through carefully designed experiments, shows the advantage of one particular approach: augmenting the observations with binary variables indicating missingness.

My opinion is that the paper is a good contribution, acting as a proof of concept for this methodology using synthetic examples. I'd like to see a comparison with existing methods, such as multiple imputation, on real data. But I'm happy for this to be done as future work, as the authors describe in the conclusion.

Comments and suggested changes are listed below. I think these should all be easy to address.

Major comments

* Algorithm 1 uses an online learning approach. Every training step generates a batch of data, which is then discarded. The authors should justify why they prefer this to an offline approach of generating the training data first and reusing it across multiple training epochs, which is discussed as an option in the Bayesflow paper and seems more efficient to me.

* Line 63: "Our presented approach is agnostic of the underlying missingness mechanism". This is not correct, as you later assume missingness indicators are sampled from π(τ), which is the definition of MCAR. The methodology could be extended to MAR and MNAR by using π(τ|x), but this is not done in the paper. Also application to MNAR would be challenging (as usual in missing data problems) because it would require correctly specifying the distribution of missingness indicators given unobserved data values.

* I liked the observation in Section 3.2 that your approach can improve on BayesFlow even outside the context of missing data as it's usually conceived. That is, it improves convergence for data of variable length. You could consider highlighting this as a secondary contribution of the paper in the abstract/introduction.

* NPE methods can be inaccurate under misspecified models (see for instance https://arxiv.org/abs/2210.06564). Adding an extra model for missingness seems to increase the potential for misspecification, especially in situations where MNAR is possible. I think addressing this issue is beyond the scope of this paper, but it would be good to add some discussion of it.

Minor comments

* Line 43: "Possible reasons exist plenty" - is this a typo?

* Line 76: "imputation... holds no conceptual advantage over approaches discarding missing data". I don't think this is correct in general. For instance see the discussion on the limitations of deletion methods in Chapter 1 of "Flexible Imputation of Missing Data" by van Buuren. I suspect the authors may be referring to the case of likelihood-based inference under ignorable missingness.

* Line 121: I think the equation involving NA implicity assumes that 0*NA=0. It would be worth pointing this out. Often the convention is that 0*NA results in NA. For instance the R language does this.

* Line 128: "Simply dropping them from the data vector is no solution." This sounds like the basis of deletion methods, which are a solution to the problem and sometimes work well. Perhaps the authors mean that this solution wouldn't work **in the context of amortized inference**, as they argue in Section 1 of the supplement.

* Section 3: I couldn't find a description of what type of cINN is used, only the number of layers.

* Line 192: You say the true posterior is the distribution given only the observed data. It's worth noting that the reason this is true is that you assume ignorable missingness.

* Figure 3: The bottom row might be more easily interpretable if you used the same y-axis in all plots.

* Line 247: "This gives a mathematically inexact approximation of the loss function, as its true value is only approximated as an average over iterations, but the network parameters are updated after each iteration." I didn't fully understand this sentence. In particular stochastic gradient methods always use inexact approximations of the loss. I think the key point here is that fixing the number of observations within a batch means the loss estimate (1) remains unbiased but (2) has higher variance (see https://en.wikipedia.org/wiki/Law_of_total_variance).

* Supplement line 63: I think the last π(xobs) on this line should have a subscript "avai".

* Supplement, just after line 67: "the missingness pattern contains information about the parameters beyond the complete data". I'm not a missing data expert, but this seems like an unusual case. It's typically to assume that π(τ|x,ψ) i.e. missingness pattern can depend on the full data and parameters ψ which are separate from the model parameters θ (in Bayesian setting typically ψ,θ are assumed to be a priori independent). Assuming also a direct dependence of τ on θ is mathematically possible, but I think the authors should make a case for why this would be a useful model.

**Have the authors made all data and (if applicable) computational code underlying the findings in their manuscript fully available?**

Reviewer #1: None

PLOS authors have the option to publish the peer review history of their article (what does this mean?). If published, this will include your full peer review and any attached files.

Reviewer #1: No

Figure Files:

Data Requirements:

Reproducibility:

References:

---

## [Decision Letter · Decision Letter 1]

21 May 2024

Dear Mr. Schälte,

We are pleased to inform you that your manuscript 'Missing data in amortized simulation-based neural posterior estimation' has been provisionally accepted for publication in PLOS Computational Biology.

Best regards,

James R. Faeder

Academic Editor

PLOS Computational Biology

Pedro Mendes

Section Editor

PLOS Computational Biology

Reviewer's Responses to Questions

**Comments to the Authors:**

Reviewer #1: I'm happy that the authors have addressed all my comments from the previous round. I also like the presentation of the previous and revised content in the response letter.

One minor point to optionally address before publication is about "discarding missing data". In the original version I thought this referred to deletion methods, but from the response letter I think this is not exactly the meaning that was intended. (I suspect the intended meaning is things like likelihood methods using only the observed data under the assumption of ignorable missingness.) You could emphasise further in the text that there is a distinction between "discarding" and "deletion". But this is an optional suggestion as your new text does already explain this.

**Have the authors made all data and (if applicable) computational code underlying the findings in their manuscript fully available?**

Reviewer #1: Yes

PLOS authors have the option to publish the peer review history of their article (what does this mean?). If published, this will include your full peer review and any attached files.

Reviewer #1: No

---

## [Editor Report · Acceptance letter]

29 May 2024

PCOMPBIOL-D-23-01114R1 

Missing data in amortized simulation-based neural posterior estimation

Dear Dr Schälte,

I am pleased to inform you that your manuscript has been formally accepted for publication in PLOS Computational Biology. Your manuscript is now with our production department and you will be notified of the publication date in due course.

With kind regards,

Zsofia Freund
